# Plasmid DNA-Based Alphavirus Vaccines

**DOI:** 10.3390/vaccines7010029

**Published:** 2019-03-08

**Authors:** Kenneth Lundstrom

**Affiliations:** PanTherapeutics, 1095 Lutry, Switzerland; lundstromkenneth@gmail.com; Tel.: +41-79-776-6351

**Keywords:** alphaviruses, layered RNA/DNA vectors, DNA vaccines, RNA replicons, recombinant particles, tumor regression, protection against tumor challenges and infectious agents

## Abstract

Alphaviruses have been engineered as vectors for high-level transgene expression. Originally, alphavirus-based vectors were applied as recombinant replication-deficient particles, subjected to expression studies in mammalian and non-mammalian cell lines, primary cell cultures, and in vivo. However, vector engineering has expanded the application range to plasmid DNA-based delivery and expression. Immunization studies with DNA-based alphavirus vectors have demonstrated tumor regression and protection against challenges with infectious agents and tumor cells in animal tumor models. The presence of the RNA replicon genes responsible for extensive RNA replication in the RNA/DNA layered alphavirus vectors provides superior transgene expression in comparison to conventional plasmid DNA-based expression. Immunization with alphavirus DNA vectors revealed that 1000-fold less DNA was required to elicit similar immune responses compared to conventional plasmid DNA. In addition to DNA-based delivery, immunization with recombinant alphavirus particles and RNA replicons has demonstrated efficacy in providing protection against lethal challenges by infectious agents and tumor cells.

## 1. Introduction

The classic approach for the development of vaccines for infectious diseases has comprised of immunization with live attenuated or inactivated agents [1]. The introduction of genetic engineering expanded the approaches of vaccine development to the application of recombinantly expressed antigens and immunogens as immunization agents [2]. Both viral and non-viral vectors expressing surface proteins and antigens have been used for immunization, first in animal models followed by human clinical trials [3]. Taking this approach has elicited strong humoral and cellular immune responses and has provided protection against challenges with lethal doses of infectious agents [4]. Similarly, recombinantly expressed tumor antigens and tumor cell proteins have elicited immune responses in vaccinated animals and provided protection against challenges with tumor cells [5].

The standard procedure for non-viral vector-based immunization involves the application of conventional DNA plasmids for the expression of the antigen in question [6]. Various approaches to improve the efficacy of delivery and the expression of antigens include polymer and liposome-based coating of plasmid vectors [7,8]. DNA delivery based on both microparticles and nanoparticles has provided promising strategies for vaccine development. Microparticle systems promote the passive targeting of antigen presenting cells (APCs) through size exclusion and supports sustained DNA presentation to cells through the degradation and release of encapsulated vaccines [7]. On the other hand, nanoparticle encapsulation provides increased internalization, enhanced transfection efficiency, and improved uptake across mucosal surfaces. Appropriate biomaterial selection can enhance immune stimulation and activation through triggering innate immune response receptors [7]. Moreover, nanoparticle-based delivery can target DNA to professional APCs. Encapsulation also adds flexibility to administration routes generating systemic and mucosal immunity resulting in more effective humoral and cellular protective immune responses.

One alternative has been to apply alphavirus-based vectors, which due to the presence of the alphavirus replicon provides a self-amplifying mechanism generating substantial gene amplification and thereby enhanced expression of the gene of interest. The increased expression levels relate to improved immune responses, but also allows the potential use of reduced quantities of plasmid DNA for vaccinations. Although the focus in this review concerns DNA-based genetic antigen preparations, a short presentation of application of alphavirus RNA replicons and alphavirus replicon particles is included. The basics of the self-amplifying replicon function is briefly described below.

## 2. Alphavirus Vectors

Alphaviruses are single stranded RNA viruses possessing a positive strand polarity [9]. The genome is encapsulated in a capsid protein structure covered by a membrane protein envelope structure. After the release of the alphavirus RNA genome in infected cells, the non-structural alphavirus proteins (nsP1-4) forms the RNA replicase complex responsible for extensive RNA replication. In expression vectors, which were first engineered for RNA replicon and replicon particle delivery, the alphavirus structural genes were replaced by the foreign gene of interest [10]. This approach required the in vitro transcription of RNA from a plasmid DNA construct, which then was directly transfected into host cells for immediate transgene expression. Alternatively, co-transfection of in vitro transcribed RNA from an alphavirus vector carrying the alphavirus structural genes allowed packaging of replication-deficient recombinant alphavirus particles. These so-called “suicide particles” are capable of one round of infection of a broad range of host cells generating high levels of transgene expression.

To be able to use alphavirus-based plasmid DNA vectors for direct immunization, a mammalian host cell compatible eukaryotic RNA polymerase II type promoter such as CMV was engineered upstream of the replicon genes [11]. DNA-based alphavirus vectors provide high biosafety levels with no risk of production of new viral progeny, but still generating high levels of transgene expression due to the presence of the alphavirus replicon. However, the host cell range is dependent on the efficacy of available transfection methods. Another issue related to plasmid DNA delivery concerns the improvement of transfer to the nucleus by the introduction of nuclear localization signals (NLS) in the vector [12].

## 3. Immunization with Alphavirus Vectors

As described above, alphavirus vectors have been utilized for vaccine development as recombinant viral particles, RNA replicons and plasmid DNA [10,11]. As the main focus here is on DNA-based vaccines, immunization studies based of recombinant alphavirus particles and alphavirus RNA replicons are only described briefly.

### 3.1. DNA-Based Immunization

Alphavirus-based DNA plasmids have been frequently used for immunization studies in animal models targeting infectious agents and different types of cancers (Table 1). For instance, a Sindbis virus (SIN) DNA vector expressing the herpes simplex virus type 1 glycoprotein B (HSV-1-gB) elicited a broad spectrum of immune responses including virus-specific antibodies and cytotoxic T cells in mice [13]. Furthermore, a single intramuscular immunization with SIN-HSV-1-gB protected mice from lethal challenges with HSV-1. In another study, a Semliki Forest virus (SFV) DNA vector expressing the bovine viral diarrhea virus (BVDV) p80 (NS3) was evaluated in BALB/c mice [14]. The administration of SFV-BVDV p80 DNA into the quadricep muscles of mice generated statistically significant cytotoxic T-lymphocyte (CTL) activity and cell mediated immune (CMI) responses against cytopathic and noncytopathic BVDV. Related to measles virus (MV), SIN DNA vectors expressing the MV hemagglutinin (pMSIN-H) and fusion protein (pMSINH-FdU) were administered either alone or boosted with a live measles virus vaccine in cotton rats [15]. The study demonstrated that neutralizing antibodies, mucosal and systemic antibody-secreting cells, memory B cells, and interferon-γ (IFN-γ)-secreting T cells were obtained after priming, further enhanced after boosting.

Protection against pulmonary measles was achieved after immunization with pMSIN-H, whereas pMSINH-FdU provided protection only after boosting with a live measles virus vaccine. In another approach, an SFV DNA vector was compared to a recombinant adenovirus expressing the classical swine fever virus (CSFV) E2 glycoprotein in pigs [16]. Significantly higher titers of CSFV-specific neutralizing antibodies were obtained after a pSFV1CS-E2/rAdV-E2 heterologous prime-boost immunization strategy compared to double immunizations with rAdV-E2 alone. Moreover, the heterologous prime-boost immunization regimen prevented viremia and clinical symptoms in pigs. In contrast, these symptoms were seen in one of five pigs vaccinated with rAdV-E2 alone. Related to HIV vaccines, an SFV DNA plasmid and a poxvirus Ankara (MVA) vector expressing an HIV Env and a Gag-Pol-Nef fusion protein were subjected to a prime-boost study [17]. It was revealed that efficient priming of HIV-specific T cell and IgG responses was achieved with a low dose of 0.2 µg SFV DNA and the priming effect seemed to relate to the number of prime administrations rather than dose. In another prime-boost study, four novel alphavirus DNA replicon vectors were engineered to express structural Core-E1-E2 or nonstructural p7-NS2-NS3 hepatitis C virus (HCV) [18]. Prime immunization with alphavirus DNA-HCV vectors followed by a heterologous boost with a vaccinia virus expressing the nearly full-length HCV genome (MVA-HCV) elicited long-lasting HCV-specific CD4^+^ and CD8^+^ T cell responses in mice presenting a promising approach for prophylactic and therapeutic HCV vaccine development. Moreover, alphavirus DNA vectors were subjected to the expression of the Ebola virus (EBOV) glycoprotein (GP) gene alone or together with the EBOV VP40 gene of Sudan or Zaire EBOV strains [19]. Both binding and neutralizing antibodies were detected in immunized mice. The alphavirus-based DNA vaccine showed superior immunogenicity in comparison to recombinant MVA vaccines. In another study, the co-expression of EBOV GP and VP40 elicited significantly higher antibody levels than for immunization with GP or VP40 alone [20]. SFV-DNA EBOV GP and VP40 co-vaccination induced EBOV-specific humoral and cellular immune responses in mice [20].

In the context of *Mycobacterium tuberculosis*, a SIN DNA vector expressing the p85 antigen (Ag85) was highly immunogenic in mice and provided enhanced long-term protection against challenges with *M. tuberculosis* [21]. In another study, the alphavirus-based Venezuelan equine encephalitis virus (VEE) DNA vector expressing a fusion of the *M. tuberculosis* antigens α-crystallin (Acr) and Ag85B named Vrep-Acr/Ag85B was evaluated in a mouse model of pulmonary tuberculosis [22]. Immunization studies elicited antigen-specific CD4^+^ and CD8^+^ T cell responses, which persisted for at least ten weeks and also induced T cell responses in lung tissues. Moreover, bacterial growth was inhibited in lungs and spleen after aerosol challenges with *M. tuberculosis*. Related to toxoplasmosis, the *Toxoplasma gondii* nucleoside triphosphate hydrolase-II (TgNTPase-II) gene expressed from an SFV DNA vector was intramuscularly delivered to mice [23]. Specific humoral responses were obtained as well as cellular immune responses associated with high levels of IFN-γ, IL-2, and IL-10 cytokines and low levels of IL-4. Partial protection against acute infection with the virulent RH strain and chronic infection with the PRU cyst strain of *T. gondii* was obtained in immunized mice.

Related to toxins, alphavirus DNA vectors expressing the Hc gene of botulinum neurotoxin serotype A (BoNT/A) demonstrated specific antibody and lymphoproliferative responses in immunized BALB/c mice [24]. Co-delivery or co-expression of granulocyte-macrophage colony-stimulating factor (GM-CSF) enhanced the immunogenicity and survival rates in immunized mice were significantly prolonged after challenges with BoNT/A. Furthermore, co-immunization with aluminum phosphate adjuvant improved the survival.

In the context of cancer, an SFV DNA vector expressing the human papilloma virus type 16 (HPV-16) E7 protein as a fusion protein with the *M. tuberculosis* heat shock protein 70 (Hsp70) elicited significantly higher E7-specific T cell-mediated immune responses in comparison to E7 expressed alone in mice [25]. Moreover, the E7/Hsp70 fusion construct showed superior potency against established E7-expressing metastatic tumors. In another study on HPV, the SFV based DNA encoding the HPV E6 and E7 antigens was subjected to intradermal administration followed by electroporation, which provided effective and therapeutic anti-tumor activity resulting in approximately 85% tumor-free mice [26]. Related to breast cancer, the HER2/neu gene was targeted due to its role in increased metastasis and poor prognosis [27]. Intramuscular administration of SIN-neu DNA elicited strong antibody responses against the A2L2 mouse breast cancer cell line expressing neu. Moreover, challenges with A2L2 cells reduced tumor incidence and tumor mass in immunized mice. Intradermal vaccination required 80% less SIN-neu DNA to reach the same efficacy compared to intramuscular administration. Furthermore, the vaccination protected against development of spontaneous breast tumors and reduction in metastasis from HER2/neu expressing tumors. In another study, mice injected in the mammary fat pad with A2L2 tumor cells were evaluated for the combination treatment of SIN-neu DNA and chemotherapy [28]. Neither immunization with SIN-neu DNA nor chemotherapy with doxorubicin or paclitaxel alone reduced tumor growth. In contrast, chemotherapy followed by vaccination with SIN-neu DNA reduced tumor growth significantly. In another study, the effect of SIN-DNA immunizations was evaluated in a solid mammary tumor model and a lung metastasis model [29]. When mice were immunized with SIN-neu DNA or an Adenovirus (Ad-neu) vector prior to challenges with A2L2 tumor cells, tumor growth was significantly inhibited. In contrast, vaccination two days after tumor cell challenges was ineffective. However, in a regimen with SIN-neu DNA priming and Ad-neu boosting, significantly prolonged survival of mice was observed.

In an immunotherapy approach SIN-DNA expressing the self/tumor antigen tyrosine-related protein-1 (TRP1) was demonstrated to activate innate immune pathways providing improved immunization efficacy of naked DNA [30]. Related to melanoma, the melanoma cell adhesion molecule/MCAM/MUC18) was expressed from a SIN DNA plasmid (SIN-MUC18) and mice were vaccinated against B16F10 mouse melanoma cells [31]. The immunization provided protection of mice from lethal challenges with melanoma expressing mouse MUC18 in both primary and metastatic tumor models. In the context of brain tumors, immunization with SIN DNA expressing human gp100 and interleukin-18 (IL-18) enhanced both protective and therapeutic effects on malignant brain tumors [33]. The anti-tumor and protective effects were mediated by both CD4^+^/CD8^+^ T cells and IFN-γ and the survival rate was significantly improved in mice with implanted B16 tumors. The synergistic approach of targeting tumor cells and angiogenesis was simultaneously executed by co-immunization studies with an SFV DNA replicon vector carrying 1-4 domains of murine vascular epidermal growth factor receptor-2 (VEGFR2) and IL-12 and another SFV DNA replicon expressing the survivin and β-hCG antigens [32]. The combined vaccines elicited strong humoral and cellular immune responses against survivin, β-hCG and VEGFR2, inhibited tumor growth and prolonged survival in a B16 melanoma mouse model.

### 3.2. Recombinant Viral Particles

Numerous immunization studies conducted with recombinant alphavirus replicon particles have been described previously [34] and as the focus on this review is on DNA-based alphavirus vectors, only two examples of comparative studies on replicon particles and DNA vectors are presented here. In this context, a study on the immunogenicity and protective efficacy of DNA-based SIN and recombinant SIN particles expressing the medium (M) or small (S) gene segments of the Seoul virus (SEOV) was conducted in Syrian hamsters [35]. Both DNA-SIN and recombinant SIN particles elicited anti-SEOV immune responses and protection against SEOV challenges was observed for all animals vaccinated with SEOV-M, but only for a small number immunized with SEOV-S. Furthermore, the study revealed that hamsters immunized with SIN-DNA developed neutralizing antibodies faster and at higher titers compared to SIN replicon particle-based delivery.

In another study, recombinant SFV particles and RNA replicons were applied for expression of the HIV-1C gag, env, and polRT genes [36]. Immunization of mice elicited significant antigen-specific IFN-γ T cell responses. Moreover, SFV-based Gag and Env expression generated TNF-α secreting CD4^+^ and CD8^+^ T-cells and IL-2 secreting T cells, respectively. In this study, superior immunogenicity was obtained for SFV particle administration in comparison to RNA replicon delivery.

### 3.3. RNA-Based Delivery

Similar to recombinant alphavirus particle delivery, RNA replicon administration has proven efficient in vaccine development [34]. For example, a single intramuscular injection of 0.1 µg SFV-LacZ replicon RNA generated antigen-specific antibody and CD8^+^ T cell responses in immunized mice [37]. Immunization with SFV-LacZ RNA prior to challenges by colon tumors provided protection in mice. Moreover, the therapeutic vaccination of animals with pre-existing tumors resulted in prolonged survival. Interestingly, the levels of antigen production for RNA replicons in vitro were not significantly higher than those observed for conventional DNA vaccines, but in vivo the enhanced efficacy correlated with a caspase-dependent apoptotic cell death. In another approach, a SIN RNA replicon expressing the rabies virus glycoprotein gene was applied for immunization studies with 10 µg of SIN-Rab-G RNA in comparison to a conventional rabies DNA vaccine and the commercial cell culture vaccine Rabipur [38]. The SIN-Rab-G RNA immunization elicited similar cellular and humoral IgG responses in comparison to the rabies DNA vaccine. Moreover, the alphavirus RNA vaccine provided similar protection to the rabies DNA vaccine against challenges with the lethal rabies virus CVS strain.

In addition to naked RNA delivery, alphavirus vectors have also been subjected to nanoparticle encapsulation procedures [39]. An in vivo expression comparison of 1 × 10^6^ IU recombinant VEE particles, 1 µg of naked replicon RNA, 1 µg of replicon RNA encapsulated in lipid nanoparticles (RNA/LNPs), 10 µg of conventional plasmid DNA, and 10 µg of replicon DNA expressing firefly luciferase was carried out in mice 7 days after bilateral intramuscular administration. The luciferase levels were similar for RNA/LNPs and VEE particles, but significantly higher than for naked replicon RNA, replicon DNA, and plasmid DNA. The immunogenicity of delivery modes was evaluated by heterologous expression of the respiratory syncytial virus fusion protein (RSV-F) after intramuscular administration. The F-specific IgG response to 1 µg RNA/LNPs was equivalent to that of 1 × 10^6^ IU of VEE particles. In contrast, plasmid DNA/LNPs at a dose of 0.1 µg and 20 µg of electroporated plasmid DNA elicited much lower IgG titers. RNA/LNPs, replicon RNA, VEE particles and an RSV-F subunit vaccine were evaluated for protection against viral challenges after intranasal RSV challenges in cotton rats. All replicon RNA vaccines protected animals for RSV challenges reducing the viral load more than 1000-fold in the lungs. The RNA/LNPs (1 µg) elicited similar responses as VEE particles. However, the recombinant F subunit vaccine formulated with alum showed the highest potency. In another study, naked RNA from SFV replicon (rSFV-NP) and poliovirus (rDELTA1-E-NP) vectors expressing the influenza type A virus nucleoprotein (NP) were intramuscularly administered in C57BL/6 mice [40]. Both rSFV-NP and rDELTA1-E-NP elicited antibodies against the influenza virus NP, but CTL responses against the immunodominant H-2D(b) epitope NP366 was only obtained with the SFV replicon RNA. Furthermore, reduced virus load was demonstrated for rSFV-NP after challenges with a mouse-adapted influenza A/PR/8/34 virus in immunized mice. The protective potential for RNA replicon immunization was similar to what has previously been achieved for plasmid DNA immunizations.

## 4. Comparison to Conventional DNA Immunization

In attempts to evaluate the feasibility of alphavirus DNA replicons as vaccine vectors, a direct comparison to conventional DNA vaccines has been an essential component. In this context, both the conventional DNA plasmid pWRG7077 and the SIN DNA replicon expressing SEOV M and S gene segments showed potential as vaccine vectors as described above [35]. However, there were substantial and to some extent surprising differences. In vitro expression levels were consistently higher from the conventional DNA vector than from the SIN DNA replicon. However, higher titers were obtained in vivo for vaccinations with SIN DNA replicons than for the conventional DNA plasmid. It has been suggested that the enhanced immune response relates to certain alphavirus vector genes promoting cell death and inducing interferon responses [41]. Moreover, as described above, immunization with SIN-TRP1 DNA broke tolerance and provided immunity to melanoma, which was not the case for conventional DNA vaccines [30]. Similarly, the long-term protection against *M. tuberculosis* obtained by immunization with SIN-Ag85 DNA was not achieved by vaccination with a conventional DNA plasmid in mice [21].

Several studies have demonstrated that in general, significantly lower doses of alphavirus DNA replicon are required to achieve the same level of response as seen for conventional DNA vaccines [14,17]. For instance, 100-fold to 1000-fold lower doses of SIN-HSV-1-gB were needed to elicit antibody responses and protection against lethal virus challenges. Moreover, a single dose of 10 ng elicited strong immune responses in mice. In the context of cervical cancer vaccines, while a conventional DNA-based vaccine failed to prevent tumor growth, immunization with a 200-fold lower equimolar dose of 0.05 µg of the SFV DNA replicon resulted in complete tumor regression in 85% of immunized mice [26]. In attempts to enhance the immune responses, the alphavirus DNA replicon vector expressing the multiclade HIV-1 T cell immunogen HIVconsv (DREP.HIVconsv) was subjected to intradermal delivery followed by in vivo electroporation and compared to the conventional DNA plasmid pTH.HIVconsv [42]. HIV-1-specific CD8^+^ T cell responses were obtained in mice with 1 µg of pTH.HIVconsv compared to only 3.2 ng of DREP.HIVconsv, which represents a 625-fold molar dose reduction. These responses could be further enhanced for both the conventional DNA plasmid and the alphavirus DNA replicon by heterologous vaccine boosts with MVA-HIVconsv and attenuated chimpanzee adenovirus ChAdV63.HIVconsv. Additionally, immunization of rhesus macaques demonstrated that application of alphavirus DNA replicon vectors allowed to reduce the dose by at least 20-fold compared to conventional plasmid DNA vectors. For this reason, the manufacturing of large batches of GMP grade material for clinical trials and marketed products is easier and more feasible. Another feature of importance related to DNA replicon vaccines is that the expression is transient and lytic, eliminating such biosafety risks as chromosomal integration and the induction of immunological tolerance [43].

## 5. Conclusions

Several studies have confirmed that alphavirus DNA replicon vectors elicit strong immune responses in vaccinated animal models targeting both infectious agents and tumor antigens. Moreover, protection against lethal challenges by viruses, bacteria, and tumor cells have also been established. In many cases, DNA replicon vaccines have proven superior to conventional DNA plasmid vaccines or at least as efficient. However, it has been confirmed that significantly lower doses of DNA replicon vaccines are needed to achieve the same immune responses and protection as for conventional DNA vaccines. In the context of alphaviruses, in addition to DNA replicon vectors, RNA replicons and recombinant alphavirus particles have also been subjected to vaccine studies. So far, there is no clear indication of which delivery format is the best and it seems more like the ranking order varies from one target to another.

Related to the biosafety of DNA vaccines, the probability of stable chromosomal integration of transfected DNA presents some concern. In this context, it was confirmed that an intramuscularly administered DNA vector expressing a luciferase reporter gene could be detected in the skeletal muscle for more than 19 months [44]. However, the DNA was only present as an extrachromosomal plasmid. When intramuscular immunization was followed by electroporation, low-level random chromosomal integration occurred, although the frequency was significantly lower than observed for spontaneous gene mutations [45]. Another study demonstrated that DNA administration into the skeletal muscles resulted in the presence of a majority of the DNA at the injection site with only minor amounts detected in other organs [46]. Moreover, no genomic plasmid DNA integration was discovered. Related to immune responses, no anti-DNA antibodies were observed after repeated intramuscular injections in primates [47]. Another issue relates to the presence of prokaryotic elements such as antibiotic resistance genes in DNA vaccines [48]. However, no transfer of such elements has been documented so far.

Another concern of alphavirus DNA replicon vaccines relates to the difficulties in transferring the strong immune responses detected in rodents to larger animals and most importantly to humans. Disappointingly, this has also been verified in clinical trials which have supported the need of dose optimization [49,50,51]. Recent studies have indicated that prime-boost strategies combining alphavirus-based vaccines with other viral-based vaccines have enhanced the immunogenicity, which is important, especially in clinical settings. Another approach briefly mentioned in this review relates to the improved delivery and stability of DNA-based vaccines through polymer and lipid encapsulation procedures. Moreover, efforts are being made to target dendritic cells in order to generate better immune responses for future vaccines. Overall, alphavirus-based DNA vaccines have the potential to provide a flexible and inexpensive alternative to current existing approaches.

## Figures and Tables

**Table 1 vaccines-07-00029-t001:** Immunization with DNA-based alphavirus vectors in animal models.

Disease	DNA Vector	Amount (µg)	Target	Model/Delivery	Response	Ref
**Infections**						
HSV	SIN	0.01–3	HSV-1-gB	mouse/i.m.	Protection against HSV-1 challenges	[13]
BVDV	SFV	100	BVDV p80	mouse/i.m.	CTL and CMI immune responses	[14]
MV	SIN	100	MV-H, MV-HFdU	rat/i.m.	Protection against MV challenges	[15]
CSFV	SFV	100	CSFV E2 + rAdV	pig/i.m.	No viremia in immunized pigs	[16]
HIV	SFV	0.2	Env, Gag-Pol-Nef	mouse/i.m.	Efficient low dose priming	[17]
HCV	SFV	0.5–50	Core-E1-E2 + MVA	mouse/i.m.	Humoral immune response	[18]
EBOV	SFV	5	EBOV GP, VP40	mouse/i.d.	Binding & neutralizing antibodies	[19]
EBOV	SFV	10	EBOV GP + VP40	mouse/i.m.	Humoral & cellular immune responses	[20]
TB	SIN	0.5–50	Ag85A	mouse/s.c.	Protection against *M. tuberculosis*	[21]
TB	VEE	20	Acr-Ag85B fusion	mouse/i.m.	Protection against *M. tuberculosis*	[22]
TP	SFV	100	TgNTPAse-II	mouse/i.m.	Protection against *T. gondii*	[23]
**Toxins**						
BoNT/A	SFV	100	BoNT/A + GM-CSF	mouse/i.m.	Prolonged survival after BoNT/A challenge	[24]
**Cancer**						
Metastasis	SFV	2	HPV E7/Hsp70	mouse/gg	Potency against metastatic tumors	[25]
Cervix CA	SFV	0.05	HPV E6-E7	mouse/i.d.	Protection against HPV	[26]
Breast CA	SIN	100	neu	mouse/i.m.	Reduced tumor incidence and tumor mass	[27]
Breast CA	SIN	100	neu + Dox & Pac	mouse/i.m.	Tumor reduction	[28]
Breast CA	SIN	100	neu + Ad-neu	mouse/i.m.	Prolonged survival in mice	[29]
Tumors	SIN	3	TRP1	mouse/gg	Activation of innate immune pathways	[30]
Melanoma	SIN	50	MUC18	mouse/i.m.	Protection against tumor challenges	[31]
Melanoma	SFV	50	VEGFR2-IL-12 +	mouse/i.m.	Prolonged survival in mice	[32]
			Survivin-βhCG Ag			
Brain CA	SIN	100	gp100, IL-18	mouse	Anti-tumor and protective effects	[33]

Acr, α-crystallin; Ad, adenovirus; Ag, antigen; BoNT/A, Botulinum neurotoxin serotype A; BVDV, bovine viral diarrhea virus; CA, cancer; CMI, cell mediated immune; CTL, cytotoxic T-lymphocyte; Dox, doxorubcin; CSFV, classical swine fever virus; EBOV, Ebola virus; gB, glycoprotein B; gg, gene gun; GM-CSF, granulocyte-macrophage colony-stimulating factor; GP, glycoprotein; H, hemagglutinin; HCV, hepatitis C virus; HPV E7, human papilloma virus E7 protein; Hsp70, heat shock protein 70 from Mycobacterium tuberculosis; HSV, herpes simplex virus; i.d., intradermal; IL-18, intereukin-18; i.m., intramuscular; MV, measles virus; MVA, modified vaccinia virus Ankara; MV-HFdU, measles virus hemagglutinin fusion protein; MUC18, melanoma cell adhesion molecule; neu, neu oncogene; Pac, paclitaxel; s.c., subcutaneous; SFV, Semliki Forest virus; SIN, Sindbis virus; TB, tuberculsois; TgNTPase-II, Toxoplasma gondii nucleoside triphosphate hydrolase-II; TP, toxoplasmosis; TRP1, tyrosine related protein-1; VEE, Venezuelan equine encephalitis virus; VEGFR2, vascular epithelial growth factor receptor-2; VP40, matrix viral protein.

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
