# Peer review of "Plasmid DNA-Based Alphavirus Vaccines"

_vaccines, 2019, doi:10.3390/vaccines7010029_

Round 1

Reviewer 1 Report

Some grammar changes/typos:

Line 29: "Taken" -->"Taking" ?

Line 68: "The application..."  This sentence is confusing and should be reworded

Line 147: "stimulating"

Line 163: "spontaneous"

Line 164: "injected"

Line 211: "pre-existing"

Author Response

Dear Reviewer 1,

Thank you for your comments. All typos you pointed out have been corrected.

Reviewer 2 Report

The manuscript reviewed plasmid DNA-based alphavirus vaccines. The manuscript is compact, but it is of help to know plasmid DNA-based alphavirus vaccines briefly.

I just comment minor points.

Page 4, lines 130-142, 151.  Mycobacterium tuberculosis (M. tuberculosis) and Toxoplasma gondi (T. gondi) should be italicized.

Page 4, line 147.  “stimulatinmg” read “stimulating”.

Page 4, line 163. “sponatneous” read “spontaneous”.

Page 5, line 179. “melanoma-expressing mouse MUC18” should be “melanoma expressing mouse MUC18”.

Page 6, line 220. “rabes” read “rabies”.

Author Response

Dear Reviewer 2,

Thank you for your comments. All typos you pointed out have been corrected.

Reviewer 3 Report

In the manuscript entitled "Plasmid DNA -based Alphavirus Vaccines", the author summarized the DNA-based Alphavirus replicon research progress in vaccine development and tumor regression.

The review is well written. Here are suggestions:

How about the safety of the DNA-based Alphavirus vaccine? The DNA may recombine with host DNA result in any side effect. The author could discuss in the review.

Author provides one table to summarize the currently status of SINV, VEEV and SFV DNA-based alphavirus vector for vaccine or tumor regression use. The table could be more informative if author could add amount of DNA used for immunization, whether need boost, route for immunization and animal model (such as BalB/c, C57Bl/6 or hamster for each study). 

Author Response

Dear Reviewer 2,

Thank you for your comments. A paragraph on the safety of DNA vaccines has been added to the Conclusion section. Moreover, an alternative Table 1 with more information has been added at the end of the manuscript for evaluation by the Editor.